# On the Efficiency Enhancement of an Actively Tunable MEMS Energy Harvesting Device

**Mortaza Aliasghary** [1] , **Saber Azizi** [2,3] , **Hadi Madinei** [3] **and Hamed Haddad Khodaparast** [3,*]

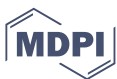

1   Electrical Engineering Department, Faculty of Industrial Technologies, Urmia University of Technology, Urmia 5716693188, Iran
2   Mechanical Engineering Department, Faculty of Renewable Energies, Urmia University of Technology, Urmia 5716693188, Iran
3   Faculty of Science and Engineering, Swansea University, Swansea SA1 8EN, UK
*   Correspondence: h.haddadkhodaparast@swansea.ac.uk

**Abstract:** In this paper, we propose an active control method to adjust the resonance frequency of a capacitive energy harvester. To this end, the resonance frequency of the harvester is tuned using an electrostatic force, which is actively controlled by a voltage source. The spring softening effect of the electrostatic force is used to accommodate the dominant frequency of the ambient mechanical vibration within the bandwidth of the resonance region. A single degree of freedom is considered, and the nonlinear equation of motion is numerically integrated over time. Using a conventional proportional–integral–derivative (PID) control mechanism, the results demonstrated that our controller could shift the resonance frequency leftward on the frequency domain and, as a result, improve the efficiency of the energy harvester, provided that the excitation frequency is lower than the resonance frequency of the energy harvester. Application of the PID controller in the resonance zone resulted in pull-in instability, adversely affecting the harvester's performance. To tackle this problem, we embedded a saturation mechanism in the path of the control signal to prevent a sudden change in motion amplitude. Outside the pull-in band, the saturation of the control signal resulted in the reduction of harvested power compared to the non-saturated signal; this is a promising improvement in the design and analysis of energy harvesting devices.

**Keywords:** capacitive energy harvesting; MEMS; PID controller; tunability; base excitation

## 1. Introduction

Providing power supplies for wireless sensor networks (WSNs) has been a popular topic in the research relating to energy harvesting devices [1–3]. The most focused long-term solution to extend the operational time of WSNs is vibration energy harvesting devices, which convert ambient vibrations into electrical energy. Today, the application of energy harvesters varies from biomedical applications [4], railway transportation [5,6], and aerospace applications [7] to energy harvesting from ocean waves [8]. Various transduction mechanisms, including piezoelectric [2,8–12], capacitive [1,13–16], and magnetostrictive [17], have been proven capable of converting the energy of the noise into electrical power. Numerous studies address the subjects of efficiency enhancement and harvesting the most energy possible [3,8–10,18]; various methods have been proposed, which include synchronous charge inversion [9], application of split proof mass [10], harmonic time-varying damping [18], application of nonlinearity by either the emergence of nonlinear resonance zones on the frequency domain or by broadening the bandwidth thanks to the softening/hardening behaviour [1,3], application of internal resonance [19,20], and a delayed feedback control scheme [8]. From the fabrication feasibility point of view, capacitive energy harvesting devices have been a central research focus thanks to their ease of fabrication. However, the fact that they require a voltage source to induce electric charges on the electrodes of the variable capacitor to start the conversion cycle [1] is a significant challenge. As mentioned, energy

harvesting enhancement and improving the efficiency of energy harvesters have drawn a lot of attention, and numerous studies have been dedicated to the topic so far. One major challenge in improving the efficiency of energy harvesters is to match the energy harvesting device in resonance with the vibration source frequency. Challa et al. [21] proposed a magnetic force resonance frequency tuning technique to change the resonance frequency of the energy harvesting beam and match the vibration source frequencies. Shi et al. [22] proposed an active self-tuning piezoelectric energy harvester, based on using a T-section mass along the cantilever beam. They demonstrated that the resonance frequency is adjusted by changing the position of a T-section mass along the cantilever beam. Lihua et al. [23] designed a vibration energy harvester, in which the resonance frequency is adjusted by changing the centroid of the harvester. Li et al. [24] proposed a nonlinear X-shaped piezoelectric energy harvester. They found that, by adjusting the structural parameters, the resonance frequency of the harvester can be tuned. Kamali et al. [25] proposed a new controller for maximum power point tracking in vibration-based energy harvesters. They demonstrated that the extremum point can be achieved by tuning the variables of the system. Staaf et al. [26] proposed a self-tuning energy harvester based on asymmetry in a system of conjoined cantilevers. They found, by varying the length of cantilevers and proof mass weights, the bandwidth of the harvester can be broadened.

We propose a capacitive energy harvester in the present study, with an actively adjustable resonance frequency. The case study is a single degree of freedom mass-spring system as a reduced-order model of a capacitive energy harvester. The electret layer between the fixed and moving plates of the capacitor is responsible for providing the voltage required for triggering the electric circuit. Once the energy harvester is excited by a harmonic mechanical noise having an excitation frequency less than the natural frequency of the energy harvester, the resonance frequency is tuned so that it is accommodated within the bandwidth of the resonator and maximizes the efficiency of the harvester accordingly.

## 2. Modelling

As illustrated in Figure 1, the case study is a single degree of freedom capacitive energy harvester with an actively tuned natural frequency. The energy harvester is exposed to a harmonic base excitation with frequency $\omega$ and amplitude $y_0$. As illustrated, the proof mass is exposed to an electret layer with a voltage $V_{DC}$ and two electrodes on either side of the electret layer, which are activated by the output of a PID controller. By applying Voltage ($V_C$) to these electrodes, the resonance frequency of the harvester can be tuned through the softening effect of the electrostatic force.

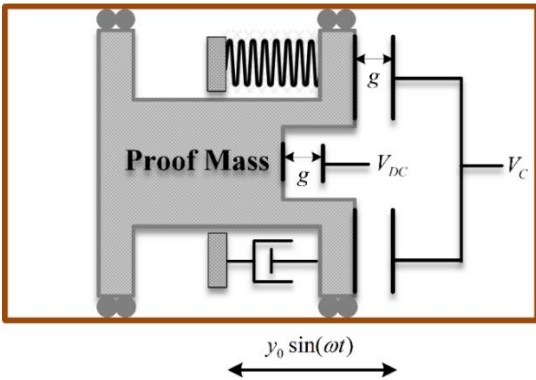

**Figure 1.** Schematics of the proposed actively tuned capacitive energy harvester.

The equation of motion along with the electrical output circuit of the energy harvester in the presence of the controller is given by Equations (1) and (2):

$$M\ddot{x} + kx + b\dot{x} = -My_b\omega^2 \sin \omega t + \frac{1}{2}\frac{\varepsilon_0 A V_C^2}{(g-x)^2} + \frac{Q^2}{2\varepsilon_0 A} \tag{1}$$

$$\frac{dQ}{dt} = \frac{V_{DC}}{R} - \frac{Q}{Rc(t)} \tag{2}$$

where $M$, $k$, and $b$ refer to the mass, stiffness, and damping coefficient of the oscillator, respectively; $y_b$ is the amplitude of the base excitation, $\omega$ is the frequency of the base excitation, $t$ denotes the time, $A$ is the surface of the proof mass exposed to the $V_{DC}$ and $V_C$, and $Q$ is the charge on the capacitors plate due to the DC electrostatic voltage; $g$ is the initial gap between the proof mass and the fixed plates of the capacitors and $R$ is the load resistance connected to the energy harvester. Equations (1) and (2) are simultaneously integrated over time to capture the time response of the system. For this, the phase space variables, including the position, velocity, and electrical current, are defined in terms of the first-order differential equations and numerically integrated over time [1].

The controller mechanism and the system are illustrated in Figure 2. Here, the reference signal, as the desired amplitude of the motion, is determined based on the motion amplitude in the resonance region.

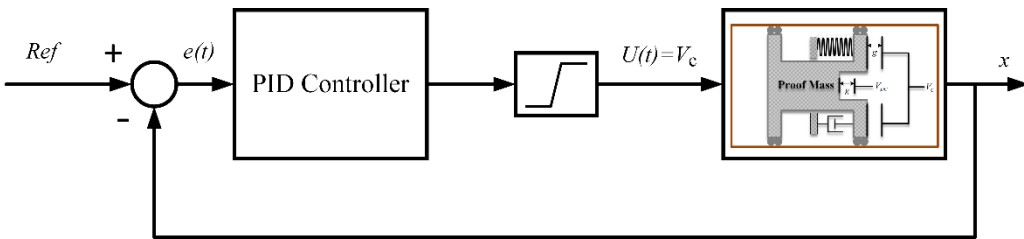

**Figure 2.** Schematics of the system along with the controller mechanism.

The output signal of the PID controller obeys the following law:

$$u(t) = k_p e(t) + k_i \int_0^t e(\tau)d\tau + k_d \frac{de(t)}{d(t)} \tag{3}$$

where $e(t)$ is the error signal, and $k_p$, $k_i$, and $k_d$ are the proportional, integral, and derivative coefficients of the control signal. The electrical charge stored in the electret layer can be obtained using the following equation:

$$Q = \frac{\varepsilon_0 A \, V_{DC}}{g - x} \tag{4}$$

By substituting Equation (4) into Equation (1), the mechanical equation of the harvester can be rewritten as:

$$M\ddot{x} + kx + b\dot{x} = -My_b\omega^2 \sin \omega t + \frac{1}{2}\frac{\varepsilon_0 A \left(V_C^2 + V_{DC}^2\right)}{(g - x)^2} \tag{5}$$

The electrostatic force term in Equation (5) can be expanded in the Taylor series as following:

$$\frac{\varepsilon_0 A \left(V_C^2 + V_{DC}^2\right)}{2(g - x)^2} = \frac{\varepsilon_0 A \left(V_C^2 + V_{DC}^2\right)}{2}\left(\frac{1}{g^2} + \frac{2x}{g^3} + \frac{3x^2}{g^4} + \frac{4x^3}{g^5} + \ldots + \text{H.O.T}\right) \tag{6}$$

Considering Taylor series expansion of the electrostatic force up to the third order, Equation (1) can be expressed as:

$$M\ddot{x} + (k - k^e)x - k_{n1}x^2 - k_{n2}x^3 + b\dot{x} = -My_b\omega^2 \sin \omega t + F_0 \tag{7}$$

where:

$$F_0 = \frac{\varepsilon_0 A \left(V_C^2 + V_{DC}^2\right)}{2g^2}, \quad k^e = \frac{2F_0}{g}, \quad k_{n1} = \frac{3F_0}{g^2}, \quad k_{n2} = \frac{4F_0}{g^3} \tag{8}$$

According to Equation (8), by increasing electrostatic voltage, the equivalent stiffness of the energy harvester decreases until the system is faced with pull-in instability. The pull-in phenomenon limits the practical application of the harvester.

## 3. Results and Discussions

The mechanical and electrical properties of the resonator are provided in Table 1.

**Table 1.** Mechanical and electrical properties of the proposed energy harvester.

| *M* | **1 μg** |
| --- | --- |
| *k* | 40 N/m |
| *b* | 0 |
| *A* | $10^4$ μm² |
| *g* | 2 μm |
| $\varepsilon_0$ | $8.8 \times 10^{-12} \left( \frac{C^2}{Nm^2} \right)$ |
| *R* | 1GΩ |
| $V_{DC}$ | 5 kv (CYPTOS as the electret layer) |

In cases where the base excitation frequency is not sufficiently accommodated in the resonance region of the energy harvester, we would neither expect the maximum possible amplitude nor the harvested power, unless we apply an appropriate electrostatic voltage to push the resonance region leftward and activate the resonance mode. Based on Equation (5), by increasing the applied DC voltage, the equivalent stiffness of the harvester decreases and, for a given applied voltage called pull-in voltage ($V_{Pull-in}$), dynamic instability occurs. The frequency response curve of the proposed energy harvester exposed to various $V_N$ is provided in Figure 3. (Considering pull-in voltage, the applied electrostatic voltage is normalized with respect to $V_{Pull-in}$ = 31,861 $v$ and is denoted by $V_N$). By increasing $V_N$, the frequency response curves move leftward along the frequency axis, and the resonance region is shifted leftward accordingly.

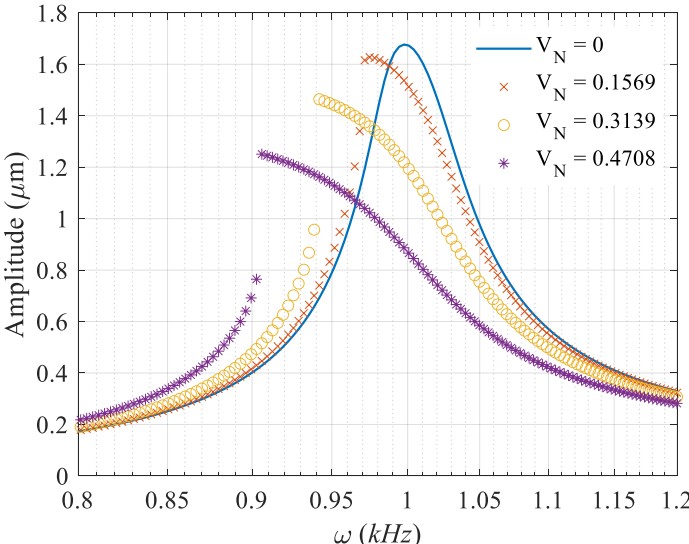

**Figure 3.** The frequency response curves with various *DC* Voltages.

Here we assume a base excitation with $y_b$ = 0.1 μm and various excitation frequencies ranging from 920 to 995 Hz, which are less than the resonance frequency of the energy harvester (1 kHz). This will satisfy the necessary condition for the controller to decrease the stiffness of the harvester. Therefore, the resonance frequency can be accommodated in the

resonance region. Figure 4 depicts the efficiency of the PID controller $\eta$, which is defined by following equation:

$$\eta = \frac{Po_c - Po}{Po} \times 100 \qquad (9)$$

where $Po_c$ and $Po$ are the harvested powers in the presence and absence of the PID controller, respectively.

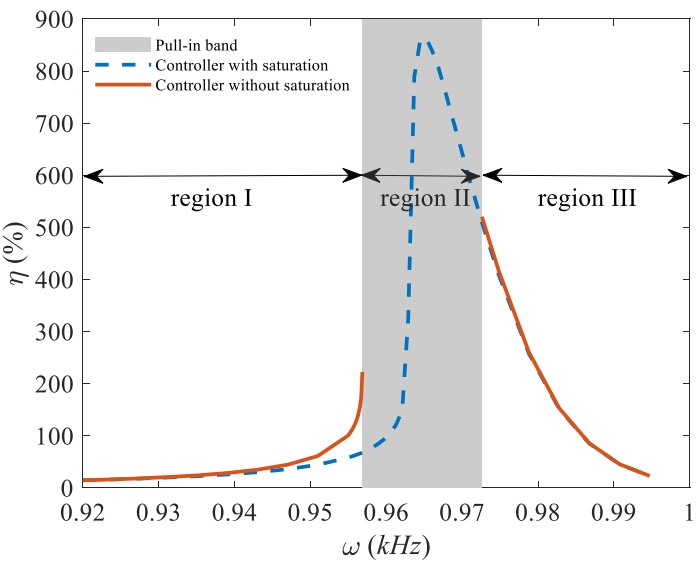

**Figure 4.** The efficiency of the controller with and without saturation.

As illustrated in Figure 4, by applying electret layer voltage $V_{DC}$, sweeping the excitation frequency and activating the PID controller, the result is an undesired pull-in band (region II) in the frequency domain due to a sudden jump in the amplitude. This is attributed to the singularity of the motion equation once the amplitude of the motion approaches $g$. To address this issue, we imposed a saturation mechanism to bound the control signal ($V_C$) and accordingly avoid a pull-in band. This is illustrated in Figure 5, where the time response, feedback voltage, and current are illustrated with the excitation frequency $\omega = 957$ Hz as the left margin of the pull-in band. In region I, as the excitation frequency approaches the margin of the pull-in band, efficiency is decreased in comparison with the non-saturated control signal; this is attributed to the limited enhancement of the amplitude of the saturated control signal. Once we pass through the margin of the pull-in band and enter region III, the system exhibits similar behaviours as in region I and the efficiency varies negligibly with the saturation; this justifies the application of non-saturated control signals in both regions I and II.

Electrostatic voltage has a softening effect on the stiffness of the resonator, and the more electrostatic voltage applied, the more the resonator softens, and pull-in instability becomes more likely. As illustrated in Figure 5, the saturation mechanism avoids pull-in by imposing a limitation on the amplitude of the control signal; however, it negligibly reduces the efficiency of the energy harvester on either side of the pull-in band. Basically, a saturated signal avoids further softening of the system and accordingly deters pull-in. Figures 6 and 7 illustrate the response of the energy harvester in regions I and III, where the application of the saturation in the control signal is not justified, and we present the corresponding response in the presence and lack of a PID controller to determine the effect of the PID controller on efficiency enhancement.

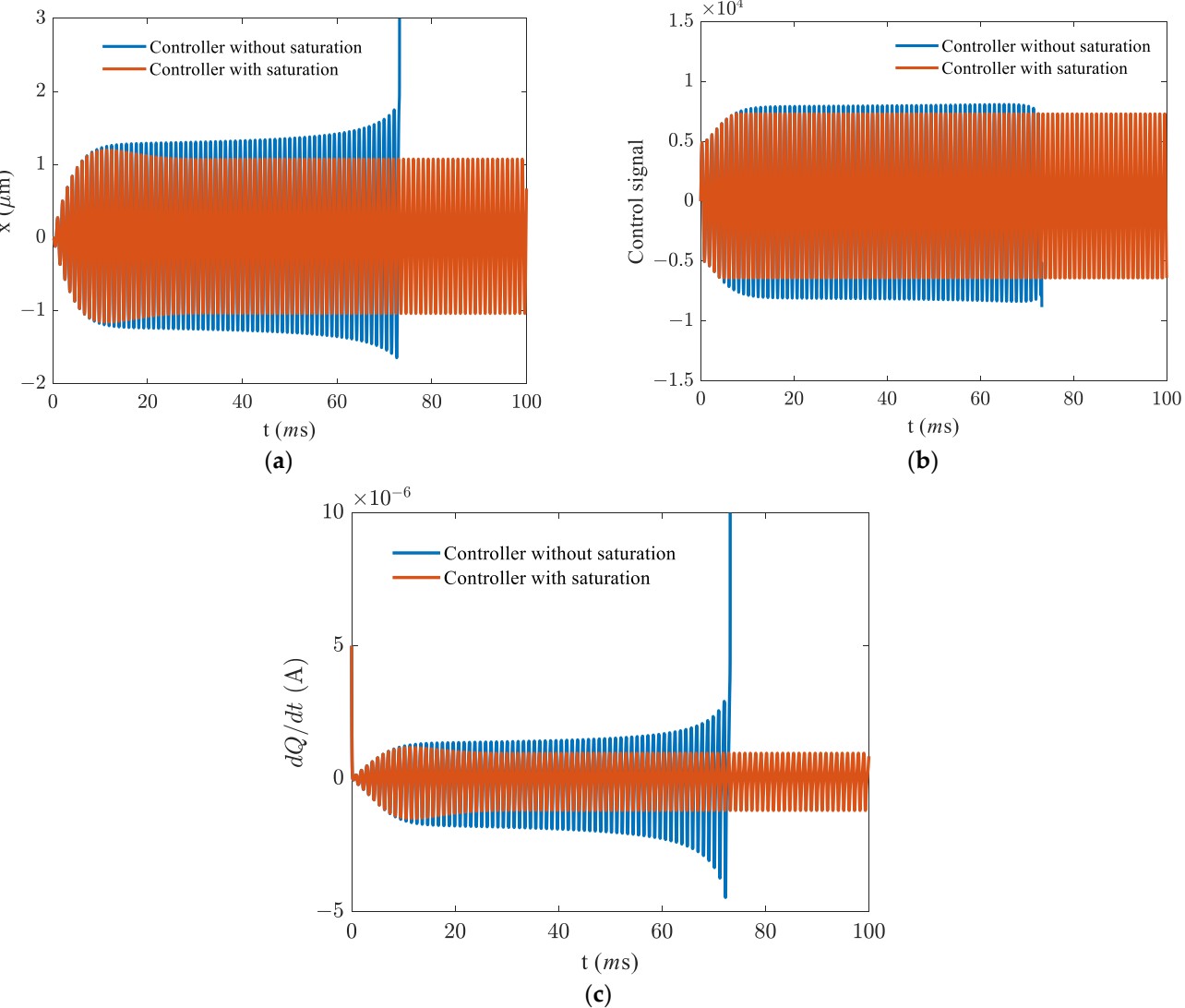

**Figure 5.** The response in the presence and absence of the saturation in the PID controller. (**a**) Time response, (**b**) feedback signal, and (**c**) current.

The base excitation frequency $\omega = 954.9$ Hz falls in region I, and the PID controller enhances efficiency up to 99.9%. We attribute this enhancement to the leftward movement of the frequency response curve (Figure 3), which activates the resonance mode by accommodating the base excitation frequency within the resonance band of the energy harvester.

Figure 8 illustrates the response of the energy harvester in region II as it is exposed to the excitation frequency $\omega = 970.8$ Hz.

Here we examine the response of the system in region III where we impose the control signal to be fed in the non-saturated form. The excitation frequency is supposed to be $\omega = 990.7$ Hz. The application of the PID controller improves the efficiency of the energy harvester by 45.3%.

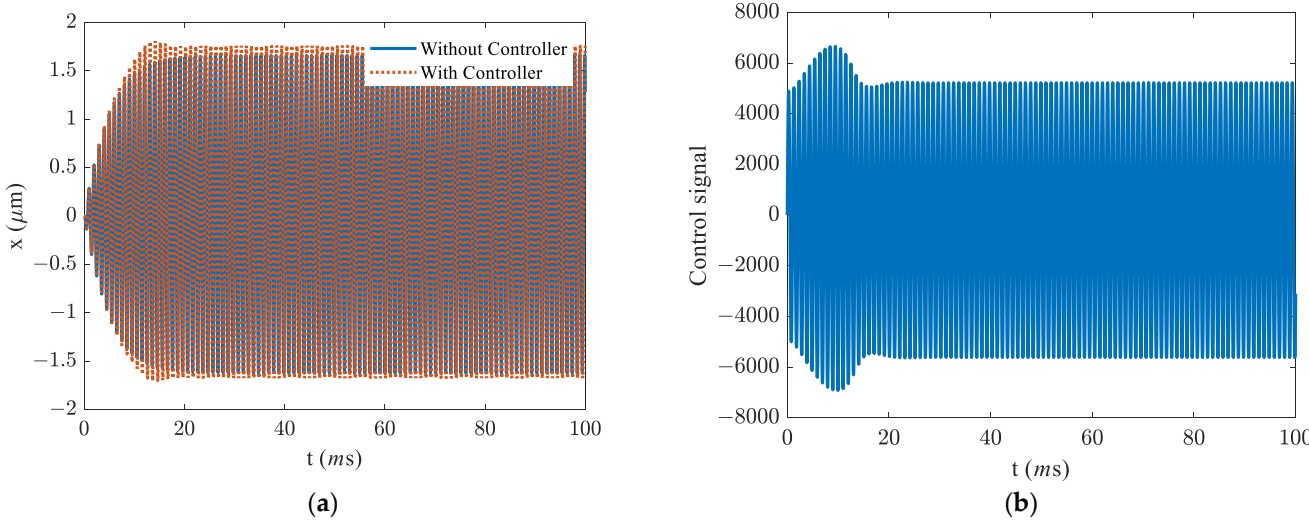

**Figure 6.** The response in region I, in the presence and absence of the saturation in the PID controller. (**a**) time response, (**b**) feedback signal, and (**c**) current.

**Figure 7.** *Cont*.

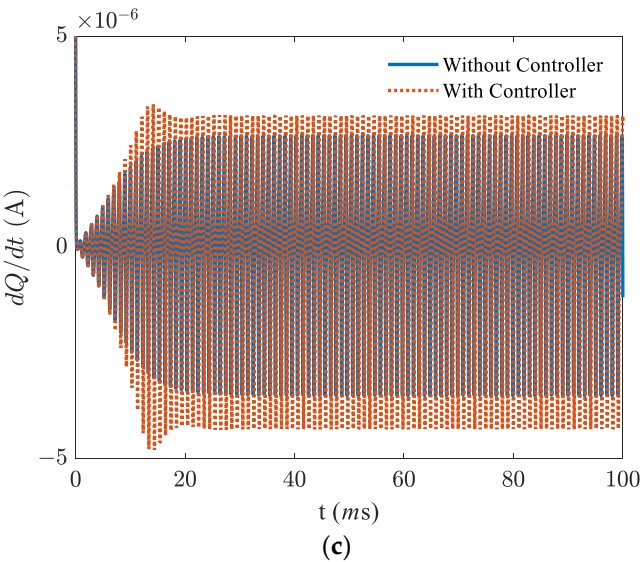

**Figure 7.** The response in region III, in the presence and absence of the saturation in the PID controller. (**a**) time response, (**b**) feedback signal, and (**c**) current.

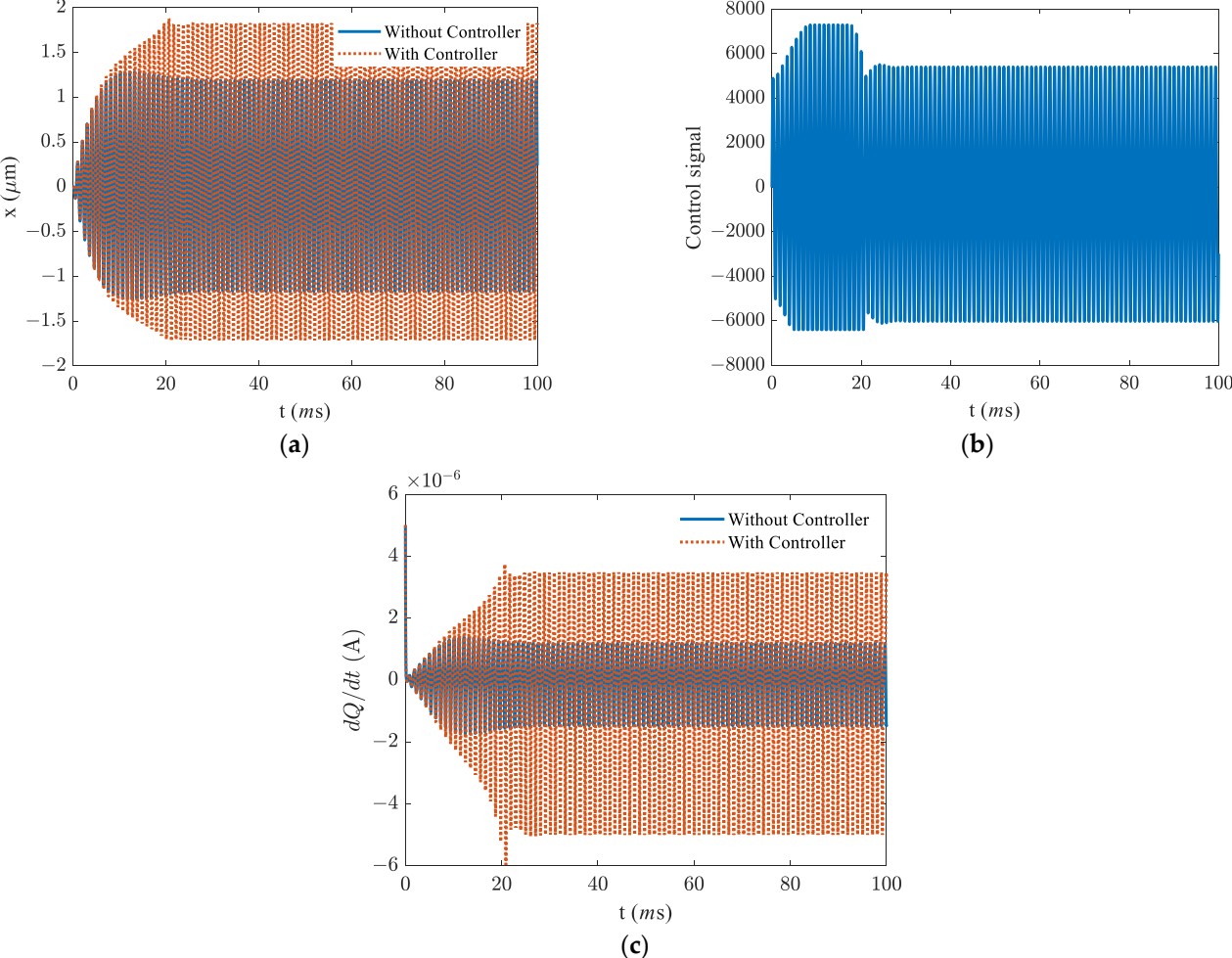

**Figure 8.** The response in region II, in the presence and absence of the saturation in the PID controller. (**a**) time response, (**b**) feedback signal, and (**c**) current.

## 4. Conclusions

In this study, we proposed an actively tuned resonance frequency capacitive energy harvester to harvest the energy from a harmonic base excitation. The equation of motion corresponding to the lumped vibrational model and the equations describing the electric circuit were numerically integrated over time to determine the time response and the power harvested. It was found that the stiffness of the harvester could be decreased by applying the control voltage. Based on this fact, a novel energy harvester capable of tuning its resonance frequency was proposed in this paper. The results demonstrated that by applying the control voltage, the resonance frequency of the harvester matches the excitation frequency. This is accomplished by imposing an actively controlled electrostatic voltage, which softens the resonator and pushes the resonance zone of the energy harvester leftward on the frequency domain accordingly. It was found that, once the PID controller is applied in the resonance zone, the effect of the electrostatic feedback voltage is amplified by the resonance, and as a result, the energy harvester undergoes pull-in instability. To avoid pull-in instability, a saturated electrostatic control signal was proposed in this paper, which increases the efficiency of the harvester. In addition, the application of the proposed tuning mechanism was examined on either side of the resonance regions. Based on the results, it was concluded that the saturation of the electrostatic feedback voltage decreases efficiency. The presented results demonstrate the potential of controlling the resonance frequency of the harvester and avoiding pull-in instability, which is promising for designing and fabricating high-efficiency capacitive energy harvesters.

**Author Contributions:** M.A. was responsible for the design of the controller and numerical simulation. S.A. was responsible for the preparation of the paper, proposing the idea and revising the paper based on the reviewer's comments. H.M. was responsible for the preparation of the manuscript and also revision of the paper. H.H.K. was responsible for the preparation od the paper and also revising the paper. All authors have read and agreed to the published version of the manuscript.

**Funding:** The paper has received no fundings.

**Conflicts of Interest:** The authors declare no conflict of interest.

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
