# Peer review of "On the Efficiency Enhancement of an Actively Tunable MEMS Energy Harvesting Device"

_vibration, doi:10.3390/vibration5030035_

Round 1
Reviewer 1 Report
The authors propose an active control method to adjust the resonance frequency of a capacitive energy harvester.
The presented work needs a major revision before being considered to be published in MPDI Vibration.
More detailed comments are as follows.
1. Authors should make a strong argument for the novelty of their work and the depth of their analysis. Generally, the paper lacks a strong and in-depth theoretical analysis.
2. The introduction is not showing the related work in the literature. The authors are advised to revise the introduction by analyzing more relevant work and showing the proposed harvester's real application.
3. The used voltage is in the order of 3kV. What is the real application of such a big applied voltage? Also, using the nondimensional Voltage V_N, what is the value of the pull-in voltage, and how did you calculate it?
4. Going from Fig.3 to Fig.4, the transition from nonlinear softening to dynamic pull-in (or escape region) is not clear. What it is the applied voltages in this case? Another analysis is needed to understand what is happening. I suggest that authors use other numerical techniques, e.g., shouting. This escape could be only due to the divergence of numerical integration. More evidence is needed to prove the findings.
5. The numerical analysis is not in-depth, and the authors should emphasize the analysis of the results.
Reviewer 2 Report
I have attached file to this email.

Round 2
Reviewer 1 Report
The authors address most of my questions.
However it is not clear again the transition to pull-in instability band. What it is the applied V_N in Figure 4? It is quite important to give this value to more follow the transition from Figure 3 to Figure 4.
Author Response
Please see the file attached.

Reviewer 2 Report
The authors have correctly made all the required changes.
Author Response
English language and style are modified.